# MODEL BREADCRUMBS: SCALING MULTI-TASK MODEL MERGING WITH SPARSE MASKS

## ABSTRACT

The rapid development of AI system development has been greatly influenced by the emergence of foundation models. The conventional approach involves fine-tuning these pre-trained foundation models for specific target tasks, resulting in a rapid spread of models fine-tuned across a diverse array of tasks. This paper introduces an innovative strategy termed *Model Breadcrumbs*, which addresses the need to merge multiple fine-tunings of the same foundation model across a spectrum of auxiliary tasks. Model Breadcrumbs consist of a sparsely defined set of weights that carve out a trajectory within the weight space of a pre-trained model, enhancing task performance when traversed. These breadcrumbs are constructed by subtracting the weights from a pre-trained model before and after fine-tuning, followed by a sparsification process that eliminates weight outliers and negligible perturbations. Our experiments demonstrate the effectiveness of combining Model Breadcrumbs to simultaneously improve performance across multiple tasks. This contribution aligns with the evolving paradigm of updatable machine learning, reminiscent of the collaborative principles underlying open-source software development, fostering a community-driven effort to reliably update machine learning models. Through extensive experimentation involving various models and tasks, we establish that integrating Model Breadcrumbs offers a straightforward, efficient, and highly effective approach for constructing multi-task models and facilitating updates to foundation models.

## 1 INTRODUCTION

In recent years, foundational models (Bommasani et al., 2021) have become instrumental, exhibiting unprecedented efficacy across multiple domains. These models are characterized by their extensive scale, generality, and capacity to learn and generalize knowledge from vast datasets, offering promising solutions to a diverse range of problems. The inherent ability of foundational models to be fine-tuned has led to advancements in natural language understanding (NLP) (Radford et al., 2018; 2019; Devlin et al., 2018; Liu et al., 2019; Raffel et al., 2020; Lewis et al., 2019), computer vision (Radford et al., 2021; Ramesh et al., 2021; Luo et al., 2020; Kim et al., 2021; Cho et al., 2021), and other related fields (Rives et al., 2021; Yin et al., 2020; Rothchild et al., 2021).

On one hand, the scalability of expanding foundational models to increase the tasks they can perform in practice poses a significant challenge as approaches such as joint training are limited in many practical scenarios (Cossu et al., 2022; Davari et al., 2022). In domains such as healthcare, stringent data privacy concerns often prohibit access to the underlying training data, even when the fine-tuned model on the said data is publicly accessible, rendering joint training infeasible (Asadi et al., 2023). Even in scenarios where access to training data is possible, the computational demands of simultaneous training on a multitude of tasks becomes restraining.

On the other hand, the widespread adoption of foundational models has led to a certain homogenization in the field (Bommasani et al., 2021). Both the training approach, commonly transfer learning from a popular foundational model (Oquab et al., 2014), and the model architecture itself have become standardized, typically following a few popular foundation models. This standardization has resulted in a proliferation of publicly available fine-tuned models, all sharing the same architecture (Wolf et al., 2020). However, beyond their conventional use for model inference, these

numerous fine-tuned models remain largely untapped, representing a missed opportunity (Ramé et al., 2022).

To address the challenges of scalability, practical constraints, and unlock the untapped potential of the growing pool of publicly available fine-tuned models, recent developments in neural network weight averaging techniques have gained attention (Izmailov et al., 2018; Neyshabur et al., 2020; Ramé et al., 2022; Ilharco et al., 2022a; Wortsman et al., 2022b;a; Choshen et al., 2022; Don-Yehiya et al., 2022). These approaches enable the practitioners to re-purpose and harness the increasingly valuable but underutilized resource that is the publicly available fine-tuned models.

More closely aligned with our work, Ilharco et al. (2022a) introduced Task Vectors, derived from the subtraction of a fine-tuned model and its corresponding pre-trained model. A collection of these Task Vectors can be amalgamated with their respective pre-trained model to construct a multi-task model, effectively repurposing pre-existing fine-tuned models without the need for additional training or access to the original training data. However, despite their potential, the merger of Task Vectors (Ilharco et al., 2022a) encounters limitations when confronted with a large number of tasks. This is primarily due to its reliance on hyperparameter tuning through validation set performance, a process that becomes computationally prohibitive at scale.

In response to these challenges and to capitalize on the untapped resources within the field, our paper introduces Model Breadcrumbs, a novel approach designed to tackle both scalability and hyperparameter generalization challenges. Model Breadcrumbs efficiently constructs multi-task models from pre-existing fine-tuned models (see Figure 1), overcoming the limitations faced by existing methods. We demonstrate that Model Breadcrumbs not only produces competitive multi-task models but also offers hyperparameters that generalize effectively as the number of tasks grows. In the next section, Section 2, we set the stage by examining related work that contextualizes Model Breadcrumbs. In Sections 3 and 4 we introduce and evaluate our framework. Finally, Section 5 offers insights into the scope and limitations of our proposed method. Our main contributions and takeaways are summarized below:

1. A novel approach to model merging and reusing the pre-existing fine-tuned models to build multi-task models, that often outperform their respective fine-tuned version.

2. We empirically show that our approach is robust towards hyperparameter perturbations, and generalizes as the number of tasks grows.

## 2 RELATED WORK

**Model Merging** Recent studies in the literature have explored the merging of models trained from scratch with different initializations (Ainsworth et al., 2022; Stoica et al., 2023). One of the main challenges in this type of model merging is aligning the models before the actual merger. Therefore, research in this branch primarily focuses on finding permutations between networks to bring them into alignment with a reference model, enabling the subsequent merger of the two models in weight space. Moreover, this branch typically focuses on the merging of only two models. Our work, on the other hand, distinguishes itself from this line of research, as we concentrate on the model merging of networks that share the same initialization, specifically initialized by a pre-trained model. Furthermore, our investigation extends beyond the merging of just two models, exploring the dynamics when multiple models are involved in the merger process.

Furthermore , Neyshabur et al. (2020)) highlighted the benefits of linearly interpolating two fine-tuned models originating from the same pre-trained model. They showed that this technique often yields a model that outperforms both of the original fine-tuned models. This discovery sparked subsequent investigations into the merging of fine-tuned models derived from a single foundation model, exploring its potential and practical applications.

Wortsman et al. (2022a) demonstrated that models fine-tuned on the same dataset with different hyperparameters can be combined together in a weighted average to yield an overall higher performing model. Unlike our work they did not consider merging models from different datasets and tasks. Choshen et al. (2022) merges models from multiple trained models in order to create a better pretrained model for downstream tasks. Unlike our work they do not demonstrate or study the creation of multi-task ability through the merging. Matena & Raffel (2022) considered merging of

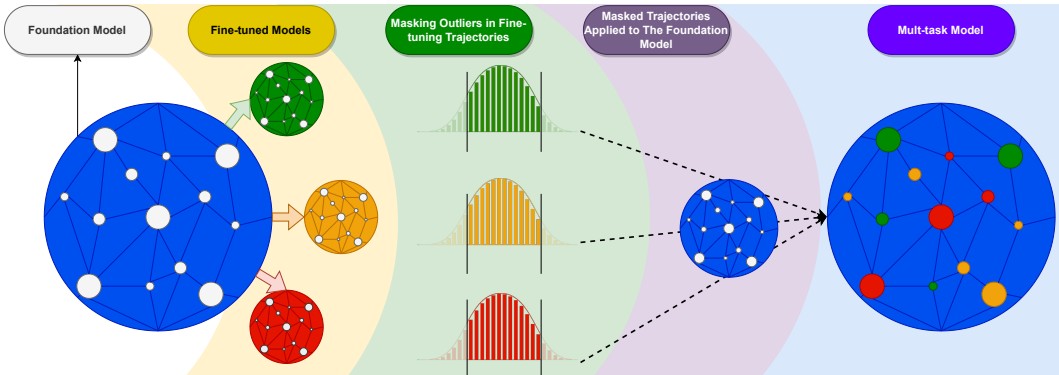

Figure 1: Method overview. We start with a foundational model that has undergone fine-tuning on various tasks. Next, we build a fine-tuning trajectory for each fine-tuned model by subtracting the pre-trained model weights from each of the fine-tuned models. We then, at each layer, apply a masking operation to the resulting trajectory, eliminating both outliers and small values. Finally, these masked trajectories are aggregated and combined with the reference pre-trained model to create a unified multi-task model.

multiple fine-tuned models originating from the same pre-trained model, trained on diverse datasets. The merger operation combines a series of fine-tuned models using a weighted average determined by the Fisher information matrix (Myung, 2003) . However, computing the Fisher information matrix, as well as finding other required hyperparameters for this approach, becomes increasingly computationally expensive as the number of models to be merged grows. Therefore, it faces challenges when applied at scale. In contrast, our approach is computationally efficient, and as we will show in Section 4, its hyperparameters exhibit the ability to generalize to a higher number of models to be merged. A closely related work to ours is the study by Ilharco et al. (2022a), in which they introduced Task Vectors. Their method involves averaging models fine-tuned for various tasks, originated from the same pre-trained model, to generate multi-task models. However, their approach necessitates a validation set for each new task, which adds complexity and computational overhead.

**Federated Learning**   The concept of initiating learning with a pre-trained model has been explored in the federated learning literature, as seen in recent works such as (Nguyen et al., 2022; Legate et al., 2023). These studies focused on a single downstream task where data is distributed across multiple clients. In their approach, each client periodically aggregates models during the training process. It's important to note that this differs from our approach, which deals with multi-task learning involving multiple downstream tasks rather than a single task distributed across clients.

## 3   MODEL BREADCRUMBS FRAMEWORK

The Model Breadcrumbs framework is designed to enable the construction of multi-task models from pre-existing fine-tuned foundation models without the need for further training. The core concept revolves around the identification and extraction of valuable knowledge from these models by navigating through their weight spaces. In this section, we provide an overview of obtaining and merging the Model Breadcrumbs.

To initiate the creation of Model Breadcrumbs, we commence with a pre-trained foundation model and subsequently fine-tune it on multiple auxiliary tasks. Denoting the weights of the foundation model as $\theta$, after fine-tuning on a specific task $t$, the weights are transformed into $\theta'_t$. The initial step involves creating Task Vectors (Ilharco et al., 2022a) by calculating the weight differences between $\theta'_t$ and $\theta$, resulting in $\theta^d_t$. This difference encapsulates the knowledge acquired during the fine-tuning process.

$$\theta^d_t = \theta'_t - \theta \qquad (1)$$

The weight differences derived in the previous step contains both large outliers, signifying considerable deviations, and insignificantly small differences that represent minor perturbations from that of the wights of the foundation model. The presence of these extremes can hinder the effectiveness of merging multiple Task Vectors (Ilharco et al., 2022a). To address this issue, we employ a sparsification process that masks out both large outliers and small differences.

The masking operation is determined by a specific percentage of weights within each layer of $\theta_t^d$. The selection process relies on the absolute magnitudes of these weights. If a weight's absolute magnitude is excessively large relative to the remaining weights in that layer, it is subject to masking (masking $\gamma$ percent of all weights in that layer). Additionally, if a weight's absolute magnitude is relatively small, it is also subject to masking (masking $\beta$ percent of all weights in that layer). The remaining weights in $\theta_t^d$ remain unmasked. This masking procedure is represented as $m_t^{\beta,\gamma}$.

The Model Breadcrumbs are constructed by applying the mask $m_t^{\beta,\gamma}$ to the Task Vectors (Ilharco et al., 2022a). We now have a set of weight differences that define a trajectory within the weight space of the foundation model. Traversing this trajectory allows us to effectively transfer the knowledge accumulated during fine-tuning across tasks, thereby enhancing performance on multiple tasks simultaneously. For a total of $T$ tasks, we assemble a multi-task model $\theta^*$ by following the trajectories defined by the Model Breadcrumbs with a specific strength parameter $\alpha$. The formation of this multi-task model can be expressed as:

$$\theta^* = \theta + \alpha \sum_{t \in T} m_t^{\beta,\gamma}.\theta_t^d \tag{2}$$

## 4 EXPERIMENTS

In this section, we conduct a series of experiments to comprehensively evaluate the Model Breadcrumbs framework. Our experiments focus on the following key aspects: 1. **Merging Model Breadcrumbs**: We incrementally add tasks, totalling 8 in our investigation, to assess the scalability and performance of merged Model Breadcrumbs as the number of tasks increases. 2. **Generalization of Hyperparameters**: We explore how the hyperparameters introduced by Model Breadcrumbs—$\alpha$, $\beta$, and $\gamma$—generalize over the number of datasets. 3. **Effect of Scale**: We investigate the impact of the scale and complexity of the foundation models on the Model Breadcrumbs' adaptability and robustness. 4. **Ablation Study**: We study the importance of the design choices introduced by Model Breadcrumbs for successful and competitive model merging.

### 4.1 DATA, METRICS, AND MORE

Our experimental builds on Ilharco et al. (2022a). In our analysis, following Ilharco et al. (2022a) we report results in terms of normalized accuracy shown below, which is defined as the ratio between the accuracy achieved by the merged model and that attained by the fine-tuned model.

$$\text{Normalized Accuracy} = \frac{\text{Accuracy of Merged Model}}{\text{Accuracy of Fine-tuned Model}} \tag{3}$$

It is noteworthy that the fine-tuned model establishes the upper bound with a normalized accuracy value of 1. Subsequently, the concept of average normalized accuracy is introduced, representing the mean normalized accuracy across multiple tasks. We assess our findings using an extensive set of 8 datasets: Cars (Krause et al., 2013), DTD (Cimpoi et al., 2014), EuroSAT (Helber et al., 2019), GTSRB (Houben et al., 2013), MNIST (LeCun et al., 2010), RESISC45 (Cheng et al., 2017), SUN397 (Xiao et al., 2010), and SVHN (Netzer et al., 2011). For more information on the datasets see Table 1.

Using the above datasets, we first fine-tune a series of CLIP models Radford et al. (2021). In our fine-tuning process, we adopt a procedure similar to that outlined in a previous study Ilharco et al. (2022b). Specifically, we conduct fine-tuning over 2000 iterations using a batch size of 128. We set the learning rate to 1e-5 and employ a cosine annealing learning rate schedule with 200 warm-up steps. The optimization is performed using the AdamW optimizer (Loshchilov & Hutter, 2017), with

| Dataset | Training | Validation | Testing | Number of classes |
|---------|----------|------------|---------|-------------------|
| Cars | 7,330 | 814 | 8041 | 196 |
| DTD | 3,384 | 376 | 1,880 | 47 |
| EuroSAT | 21,600 | 2,700 | 2,700 | 10 |
| GTSRB | 23,976 | 2,664 | 12,630 | 43 |
| MNIST | 55,000 | 5,000 | 10,000 | 10 |
| RESISC45 | 17,010 | 1,890 | 6,300 | 45 |
| SUN397 | 17,865 | 1,985 | 19,850 | 397 |
| SVHN | 68,257 | 5,000 | 26,032 | 10 |

Table 1: Data statistics.

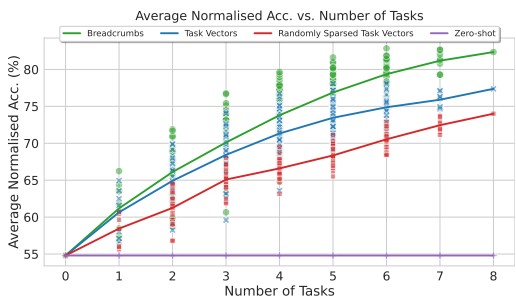

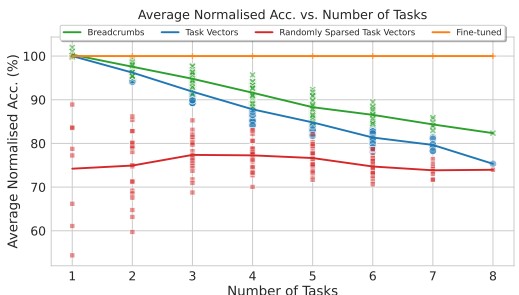

(a) At each point, evaluation is performed over all 8 tasks considered in our study.

(b) At each point, evaluation is performed only over the observed subset of tasks.

Figure 2: The solid line is the averaged normalized accuracy across all evaluation points. Each data point corresponds to an experiment involving a subset of the 8 tasks under study. Notably, it is evident that the Model Breadcrumbs (with 85% sparsity), consistently outperform the Task Vectors (Ilharco et al., 2022a). Specifically, in the experiment involving all eight tasks, the Model Breadcrumbs outperform the Task Vectors by a substantial margin of 8.33%.

a weight decay of 0.1. During the fine-tuning process, we maintain the weights of the classification layer generated by CLIP's text encoder in a frozen state. This approach ensures that we do not introduce additional learnable parameters, a strategy that has been validated in prior work (Ilharco et al., 2022b).

## 4.2 MERGING MODEL BREADCRUMBS

In this section, we investigate the scalability and performance of merged Model Breadcrumbs as we incrementally add tasks, totalling 8 in our study, as listed in Section 4.1. Merging allows us to construct multi-task models capable of excelling across multiple tasks concurrently. This versatility is valuable both in scenarios where multiple privately fine-tuned models exist and in cases where publicly available models are utilized. It enables the utilization of existing knowledge from these models without necessitating additional training or access to more training data. We compare the Model Breadcrumbs with 85% sparsity and the recently proposed Task Vectors (Ilharco et al., 2022a) in terms of their impact on the overall model's performance.

We assess all possible task subsets, amounting to a total of $2^8$ combinations, under two settings: 1. evaluation over all 8 tasks and, 2. evaluation only on the subset of tasks that have been observed. As we can see in Figure 2a merging Model Breadcrumbs results in superior multi-task models compared to the Task Vectors (Ilharco et al., 2022a). Furthermore, the performance gap between these two approaches increases as more tasks are observed, resulting in vastly superior multi-task models when more Model Breadcrumbs are available.

In Figure 2b we can see that for small task numbers the resulting merged model performs closely to that of the multiple fine-tuned models although the gap increases as more tasks are added. Model Breadcrumbs again prove to be more performance that Task Vectors (Ilharco et al., 2022a) in this setting.

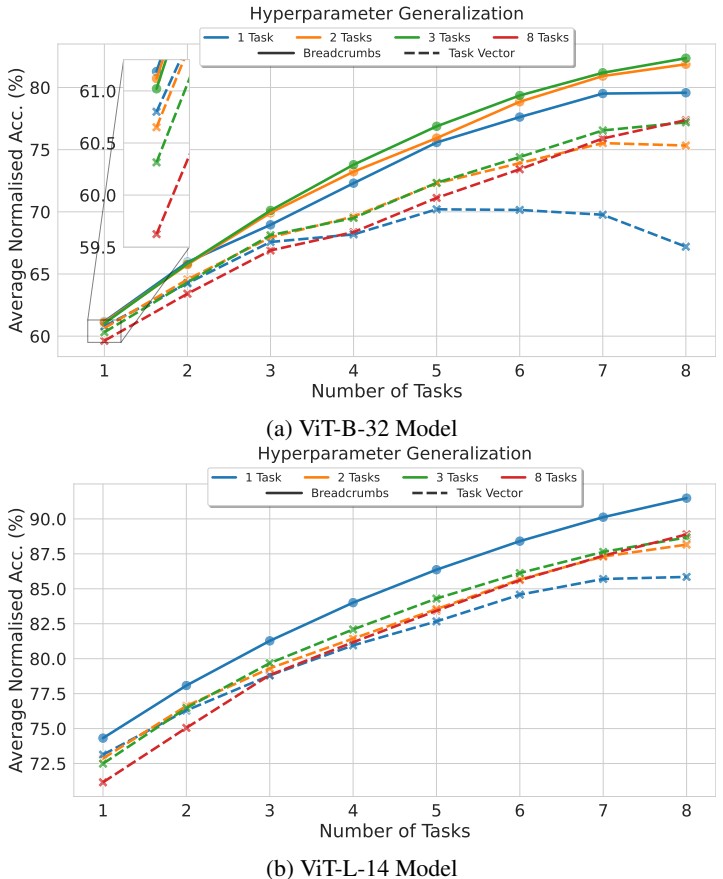

(a) ViT-B-32 Model

(b) ViT-L-14 Model

Figure 3: Validation Free Setting. For the ViT-B-32 model, we tune the hyperparameters of each method (breadcrumbs and task vectors) based on the first 1,2, or 3 tasks and add additional tasks using those hyperparameters (validation set free). Moreover, for the ViT-L-14 model, we only tune the hyperparameters for the 1 task scenario and evaluate on the additional tasks using those hyperparameters. We observe that breadcrumbs substantially outperforms task vectors in this setting.

Overall, Model Breadcrumbs consistently outperform the Task Vectors proposed by Ilharco et al. (2022a). Specifically, when evaluating these methods across the 8 tasks using their respective optimal hyper-parameters, the Task Vectors (Ilharco et al., 2022a) achieve an average normalized accuracy of 75.33%, whereas our proposed Model Breadcrumbs achieve a significantly higher accuracy of 83.00%, while having an 85% sparsity.

## 4.3 VALIDATION-FREE SETTING

In Section 4.2, we compared Model Breadcrumbs and Task Vectors (Ilharco et al., 2022a) under their respective optimal hyperparameters. These hyperparameters were fine-tuned based on model performance on the validation dataset for each subset of tasks following Ilharco et al. (2022a). However, as the number of tasks increases, the search for optimal hyperparameters becomes increasingly resource-intensive. Furthermore, the need for a validation set from each task being added can be restrictive due to privacy concerns or due to the unavailability of additional validation data. Thus we consider a new setting where hyperparamters are tuned based on a few tasks, and subsequent tasks are added using these pre-determined hyperparameters.

The results are shown in Figure 3. Remarkably, our experiments reveal that the hyperparameters of Model Breadcrumbs exhibit a high degree of generalizability. Specifically, for the ViT-B-32 model when considering scenarios involving three tasks and beyond, up to the 8-task scenario, the optimal hyperparameters remain consistent. Moreover, for the ViT-L-14 model, the hyperparameters do

not change beyond the 1 task scenario. This remarkable stability underscores the robustness and versatility of Model Breadcrumbs. We observer that on the other hand the approach of Ilharco et al. (2022a) can quickly collapse in performance.

The practical implication of this stability in hyperparameter settings is that, in practice, we can rely on a relatively small number of tasks to determine optimal hyperparameters when applying Model Breadcrumbs to diverse multi-task learning scenarios. This simplifies the implementation process, reduces the need for extensive hyperparameter tuning, and contributes to the framework's practicality and ease of use.

In contrast, Task Vectors (Ilharco et al., 2022a) do not exhibit the same level of hyperparameter stability. Consequently, this fundamental divergence between Model Breadcrumbs and Task Vectors (Ilharco et al., 2022a) underlines the substantial advantage of Model Breadcrumbs in real-world multi-task learning scenarios.

## 4.4 Effect of Scale

In this section, we explores the impact of using larger CLIP models on our analysis, comparing the performance of ViT-B-32, ViT-B-16, and ViT-L-14 models. For each model type, the optimal Model Breadcrumbs were found at 85% sparsity. As shown in Figure 4, the adoption of larger models significantly improves the performance of both our proposed Model Breadcrumbs method and the Task Vector (Ilharco et al., 2022a) baseline. Furthermore, as more tasks are introduced, the capacity to construct better-performing multi-task models grows, with larger-scale models demonstrating superior results.

Specifically, we observe in Figure 4a, when utilizing the ViT-L-14 model and considering 8 tasks, merging Model Breadcrumbs produces a single multi-task model with an average performance that reaches 91.48% of the performance achieved by employing 8 individual fine-tuned models (i.e., one per task). The shift from 8 fine-tuned models to a single multi-task model substantially reduces inference time and compute resources, accompanied by only a minor relative loss in performance. This underscores the practical advantages of our approach.

Moreover, Figure 4b highlights that the performance decline observed when merging either Model Breadcrumbs or Task Vectors (Ilharco et al., 2022a) can be significantly mitigated by adopting larger-scale models. Notably, for the ViT-L-14 model, merging Model Breadcrumbs for certain tasks can result in multi-task models that either match or surpass the performance of individual fine-tuned models. To delve deeper into this phenomenon, we conducted a closer examination of task merger for ViT-L-14, considering the two tasks scenario. Our observations reveal that:

As we can see in Figure 5, when adding pairs of tasks via Model Breadcrumbs and Task Vectors (Ilharco et al., 2022a), we observe that the merger of task vectors (Ilharco et al., 2022a) from two tasks generally leads to improved performance on both tasks, resulting in a single model that is competitive and often superior to using two specialized fine-tuned models. Furthermore, for the same task pairs, Model Breadcrumbs consistently produce multi-task models that surpass their equivalent Task Vectors (Ilharco et al., 2022a) versions. Notably, Model Breadcrumbs mergers generate a higher number of multi-task models where both tasks exceeded their respective fine-tuned accuracy levels. This highlights the potential of Model Breadcrumbs not only to maintain but also to enhance task-specific performance within a multi-task framework.

## 4.5 Ablations

In this section, we perform ablations to examine alternative design decisions within the Model Breadcrumbs method. Specifically, we explore different approaches for constructing the masking operation, namely: 1. Bottom-Weight Masking: Masking only the bottom-most smallest absolute magnitude weights per layer. 2. Top-Weight Masking: Masking only the top largest absolute magnitude weights per layer.

We compare these alternatives to the full Model Breadcrumbs approach, which encompasses both (1) and (2), as well as the Task Vectors (Ilharco et al., 2022a) method, which lacks any masking. Our goal is to assess the impact of these design choices on the performance of the merged multi-task models derived from these approaches. In our investigation, we conduct a grid search to identify

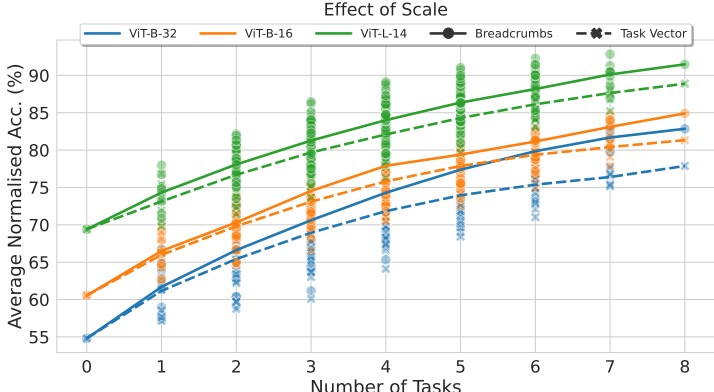

(a) At each point, evaluation is performed over all 8 tasks considered in our study.

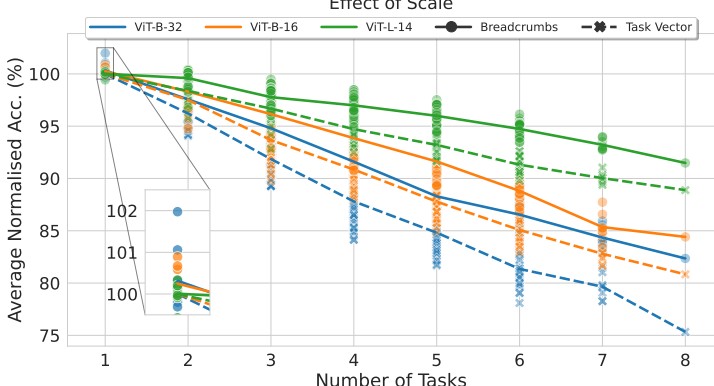

(b) At each point, evaluation is performed only over the observed subset of tasks.

Figure 4: Comparative performance analysis of Model Breadcrumbs and Task Vector (Ilharco et al., 2022a) methods across varying CLIP model scales (ViT-B-32, ViT-B-16, and ViT-L-14) as the number of tasks increases. The solid line represents the averaged normalized accuracy across all evaluation points. Each data point corresponds to an experiment involving a subset of the 8 tasks under study. Our findings highlight the potential of larger-scale models to mitigate performance degradation and, as seen in Figure 4b, the capability of Model Breadcrumbs to produce multi-task models that surpass individual fine-tuned models for specific tasks.

the optimal hyperparameters for each of the four configurations. We assess the resulting multi-task models on 8 tasks discussed in Section 4.1. The results are shown in Figure 6.

Our findings reveal two key insights: (i) both forms of weight masking, as employed in Model Breadcrumbs, are essential for achieving competitive performance. Model Breadcrumbs, which combines both bottom and top weight masking, emerges as the most effective approach. (ii) The grid search for hyperparameters within the Model Breadcrumbs approach yields a higher distribution of high-performance multi-task models compared to the other three settings. Furthermore, there is much lower variation in the overall performance distribution of the multi-task models produced by the Model Breadcrumbs. These observations underscore the robustness of Model Breadcrumbs to variations in hyperparameter settings, further enhancing its practicality and reliability in real-world applications.

## 5 CONCLUSIONS

In this paper, we introduced Model Breadcrumbs, a novel approach that addresses the challenge of constructing multi-task models from pre-existing fine-tuned foundation models. Our method effi-

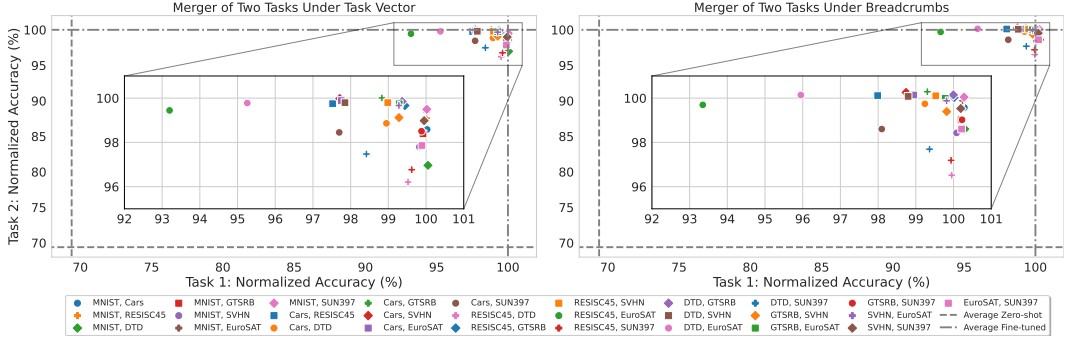

Figure 5: Comparative analysis of Model Breadcrumbs and Task Vectors (Ilharco et al., 2022a) in the merger of task pairs, revealing improved accuracy on both tasks and a higher frequency of multi-task models surpassing individual fine-tuned accuracy levels when employing Model Breadcrumbs.

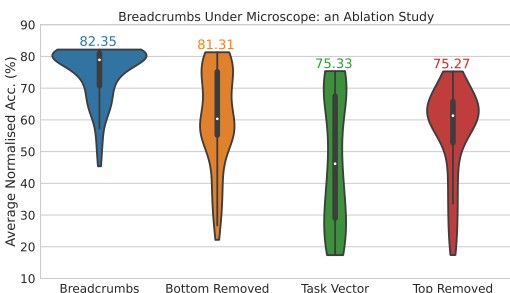

Figure 6: Performance comparison of the Model Breadcrumbs against alternative masking choices, reveals: (1) Model Breadcrumbs yields a higher distribution of high-performance multi-task models, underlining its robustness towards hyperparameter perturbations. (2) Model Breadcrumbs produces the highest performing multi-task model. The number on top of each violin indicates the performance of the highest performing model of that setting.

ciently leverages weight differences between a foundation model and its various fine-tuned versions to create guiding trajectories within the weight space of the pre-trained model, leading to high-performing multi-task models when the trajectories are traversed.

Through extensive experimentation, we have demonstrated the effectiveness of Model Breadcrumbs in simultaneously enhancing performance across multiple tasks. Notably, our approach exhibits stable and generalizable hyperparameters, simplifying its implementation and rendering it highly practical for real-world multi-task learning scenarios. Furthermore, our exploration of model scale has revealed that Model Breadcrumbs benefits from larger-scale models, closing the performance gap between merged models and individual fine-tuned models.

While Model Breadcrumbs presents a promising approach for constructing multi-task models, it does come with certain limitations. Its performance can still be affected by the quality of the fine-tuned models used as the starting point. If the fine-tuning process leads to models with poor generalization or severe overfitting, Model Breadcrumbs may inherit these issues. Future research efforts can focus on addressing the potential limitations related to the quality of fine-tuned models. Moreover, currently, Model Breadcrumbs assigns the same weight when averaging over multiple trajectories, but more sophisticated aggregation techniques could be investigated to improve performance.

In conclusion, Model Breadcrumbs offers a straightforward, efficient, and highly effective approach for constructing multi-task models, capitalizing on the abundance of the publicly available fine-tuned models derived from a select few foundation models. Additionally, it facilitates updates to foundation models, aligning with the evolving paradigm of updatable machine learning and fostering community-driven efforts for model refinement. We anticipate that our approach will contribute to the development of more efficient and scalable multi-task learning solutions in the future.

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
