# OpenReview forum: "Model Breadcrumbs: Crafting Multi-Task Models from Pre-Existing Fine-Tuned Foundation Models"
_ICLR.cc/2024/Conference — ICLR 2024 Conference Withdrawn Submission_

### Official Review · Reviewer_ffLz · 2023-10-29

**Soundness:** 2 fair
**Presentation:** 2 fair
**Contribution:** 1 poor
**Rating:** 3
**Confidence:** 4

**Summary:**

The paper presents "model breadcrumbs" as an enhanced mechanism to combine fine-tunings of a shared base model for a range of additional tasks. To create the "model breadcrumbs", the paper proposes to compute the differences between the weights of the models before and after fine-tuning and reject outliers among the differences in each layer. The resulting inlier weights are linearly combined across fine-tuned models and base model to provide enhanced generalization. The proposed method shows performance improvement over the task vectors approach used as base.

**Strengths:**

- Results show improvements with respect to the baseline (Ilharco et al., 2022a).

**Weaknesses:**

- The paper proposes a per-layer bottom and top percentile rejection of fine-tuned weights to be combined with the pre-trained model, but this proposal is not compared to any outlier-detection alternative.
- The paper excessively relies on (Ilharco et al., 2022a) throughout the text (I counted up to 3 instances in a single paragraph), resulting in a hard to read paper.

**Questions:**

- Since the paper heavily relies on a single piece of previous work, I would suggest to rewrite the conclusions to more appropriately reflect the scope of the contribution of the paper.

---

### Official Review · Reviewer_dEWb · 2023-10-30

**Soundness:** 2 fair
**Presentation:** 3 good
**Contribution:** 2 fair
**Rating:** 3
**Confidence:** 3

**Summary:**

The paper studies the problem of leveraging foundation models for multi-task learning. It introduces Model Breadcrumbs, which is a method that constructs multi-task models from existing fine-tuned models. It builds on the Task Vectors paper (Ilharco et al. 2022) and claims to resolve existing limitations--scalability and hyperparameter tuning, particularly when the number of tasks increases. More specifically, to handle outliers and insignificant weight differences, which could otherwise impair the effective merging of multiple Task Vectors, the authors propose a sparsification process employing a masking operation. The authors claim that Model Breadcrumbs yields high-performing multi-task models is robust towards hyperparameter perturbations, generalizing as the number of tasks increases.

**Strengths:**

- The motivation for the paper regarding improving the scalability and use of the growing pool of available fine-tuned models is clear and intuitive.
- The method is empirically evaluated on 8 datasets.
- The paper is overall well written and easy to follow.

**Weaknesses:**

- The method has very limited novelty over Task Vectors and is hacky. It is also not described very rigorously. The selection process for masking relies on the absolute magnitudes of the weights, but how do you decide what is "excessively large" or "relatively small"?
- There is no clear justification for the hacky masking method. Presumably, it is just having some sort of regularizing effect, but this is not explored and there are other ways that people can regularize models to improve performance.
- In addition, the experimental evaluation is quite limited, even compared to the original Task Vectors paper, which considered other benchmarks (including language tasks) as well. Additionally, the authors claim that they do not need a validation set but in fact it seems like they still do in order to tune hyperparameters on at least some of the tasks.
- Overall, the paper presents very limited new insights over the existing Task Vectors paper.

**Questions:**

See weaknesses above.

---

### Official Review · Reviewer_mobQ · 2023-10-31

**Soundness:** 3 good
**Presentation:** 2 fair
**Contribution:** 2 fair
**Rating:** 6
**Confidence:** 4

**Summary:**

This paper proposes a method to merge mutiple fine-tuned model (on mutiple tasks). The method measures the magnitude of "task vector"  which equals to fine-tuned weights - initial pretrained weights. Then mask out top $\gamma$ and tail $\beta$ percent elements in task vector according the magnitude of the task vector. Experimental results show the proposed method is helpful.

**Strengths:**

- The proposed method is intuitive.
- Experiments show the proposed weight averaging method is better than the naive weight averaging.
- Authors provide a rich literature review of model merging.

**Weaknesses:**

- The method and experiments are kinds of superficial. The idea of elimiting large outliers and small noise is very common, e.g. [1].
-  Mutiple fine-tuned non-convex neural network could be not-averagable. However, this work didn't discuss this point.


[1] Yadav, P., Tam, D., Choshen, L., Raffel, C., & Bansal, M. (2023). Resolving Interference When Merging Models. arXiv preprint arXiv:2306.01708.

**Questions:**

- I suggest to compare with or at least discuss [1]. Because [1] is close to the proposed method.
-


[1] Yadav, P., Tam, D., Choshen, L., Raffel, C., & Bansal, M. (2023). Resolving Interference When Merging Models. arXiv preprint arXiv:2306.01708.

---

### Official Review · Reviewer_VPFm · 2023-11-07

**Soundness:** 2 fair
**Presentation:** 3 good
**Contribution:** 2 fair
**Rating:** 3
**Confidence:** 4

**Summary:**

In this paper, a model merging method is proposed. The motivation lies in merging models fine-tuned for multiple tasks from a single foundation model. The proposed method is based on previous work of task vector [Ilharco et. al., 2022], and further design a pruning strategy that masks outlier model weights with small and large weights. Experimental results on benchmark datasets verify the effectiveness of the proposed method.

**Strengths:**

- The paper is well-motivated. It studies a very interesting problem: merging multiple models fine-tuned from a single foundation model into a single one. A solution of this problem would be very useful in real applications.

- The experimental results show that the proposed method is effective on a number of benchmark datasets.

**Weaknesses:**

- The proposed method is somehow a tweak of the task vector method. Thus the technical contribution is limited. More importantly, the masking strategy is more of a heuristic method. The analysis of why it works is missing in the paper. The analysis is essential to justify that the proposed method can work beyond the benchmark datasets in the experiments.

- The description of the proposed method is not quite clear. In the second paragraph of Page 4, it is said that "If a weight’s absolute
magnitude is excessively large relative to the remaining weights in that layer, it is subject to masking". Does this mean that the large weights are set to zero? Or just thrown away like dropout? The description should be made precise.

- I also suggest experimental investigation on the proportion of shared and deviated weights of models fine-tuned from a single foundation model, as well as how they are distributed (e.g. more similar in bottom layers or the opposite). I believe that more insights can be obtained from such kind of analysis.

**Questions:**

As discussed above, precise description of the masking operation is necessary.